# Oral Chagas disease outbreak by bacaba juice ingestion: A century after Carlos Chagas' discovery, the disease is still hard to manage

Eudes Alves Simões-Neto[1]*, Daniel Wagner de Castro Lima Santos[2,3], Maria Rosa Quaresma Bomfim[4], Jackson Maurício Lopes Costa[5], Amanda Ferreira Simões[6], Lucas Dias Vasconcelos[1], Domingos Carvalho Sodré[5], Ana Cleide Mineu Costa[5], Samuel Vieira Rodrigues Dumont[5], Bruna de Oliveira de Melo[4], Conceição de Maria Pedrozo e Silva de Azevedo[1]

1 Health Sciences Graduate Program, Federal University of Maranhão, São Luís, Brazil, 2 Presidente Dutra University Hospital (HU-UPD), Brazilian Company of Hospital Services (EBSERH), Federal University of Maranhão (UFMA), São Luís, Brazil, 3 IDOR, Instituto D'Or de Pesquisa e Ensino, 4 Ceuma University (UNICEUMA), São Luís, Brazil, 5 State Health Department of Maranhão (SES/MA), São Luís, Brazil, 6 Brasília University Hospital (HUB), Brazilian Company of Hospital Services (EBSERH), Brasília University (UnB), Brasília, Brazil

* eudesimoes@gmail.com, eudes.simoes@ufma.br

## Abstract

### Background

Orally transmitted acute Chagas disease (ACD) primarily affects low-visibility and low-income individuals in tropical and subtropical zones. Managing ACD remains challenging even after more than 100 years of its discovery. Its spread to non-endemic areas has made it a global health issue. The aim of this work is to demonstrate the difficulties encountered in handling a real-life situation.

### Methodology and findings

This report examines an outbreak of 39 cases of ACD due to oral transmission by bacaba juice ingestion that occurred in Pedro do Rosário, Maranhão, Brazil. A clinical and epidemiological investigation, including an entomological search, was conducted. Diagnosis criteria included positive peripheral blood smear (PBS), seroconversion of IgG, and a two-fold increase in IgG titer (laboratory criteria); and clinical findings, epidemiological exposure, and at least one positive IgG test (clinical-epidemiological criteria). In-house conventional polymerase chain reaction (PCR) was performed on 33 samples. All patients were treated with benznidazole. After 4.5 years, IgG levels were reassessed in 26 individuals. The mean age was 33.6 years, with no gender difference. The mean incubation period was 13.8 days, and the mean between symptom onset and treatment was 16.6 days. The most common symptoms were fever and lymphadenopathy (90%). Diagnostic success rates were 66.6% (laboratory criteria), 23% (clinical-epidemiological criteria), and 10.2% (high clinical suspicion despite negative tests). Test positivity rates were 69.7% (PBS), 91.4% (serology), and

**Data Availability Statement:** All relevant data are in the manuscript and its Supporting information files.

**Funding:** The author(s) received no specific funding for this work.

**Competing interests:** The authors have declared that no competing interests exist.

100% (PCR). There were no deaths. Serological cure was achieved in 34.6% of cases, and IgG titers decreased in 15.3%.

## Conclusions and significance

We encountered several barriers in managing ACD, including population vulnerability, reliance on outdated diagnostic techniques, lack of standardized molecular biology methods, and limited therapeutic options. This report underscores the importance of rapid surveillance and early treatment to prevent fatalities. We recommend the standardization of conventional PCR in diagnostic routines.

### Author summary

In this study we present an outbreak of orally transmitted acute Chagas disease (ACD), a neglected tropical disease. The source of transmission was bacaba juice served at a celebration held in a quilombola community and 39 individuals were affected. The focus of this study was clinical-epidemiological investigation and diagnostic tools. There were no deaths due to this event, probably because of the quick action of the surveillance team and the early start of treatment. We show all the difficulties facing a real-life situation (vulnerability of population affected, weaknesses regarding good food safety practices and obstacles to following what guidelines recommend), discuss the performance and feasibility of the diagnostic method used and address the global threat that Chagas disease may pose. Furthermore, we propose that the polymerase chain reaction should become standardized in diagnosing acute disease, given that this was the only method with 100% positivity, even among patients whose other tests were all negative.

## 1. Introduction

Chagas disease (CD) is a chronic systemic anthropozoonosis in the Americas caused by *Trypanosoma cruzi* (Chagas 1909) (Kinetoplastida, Trypanosomatidae), a hemoflagellate intracellular parasite discovered in 1909 [1,2]. American trypanosomiasis is classified by the World Health Organization (WHO) as one of the 20 neglected tropical diseases (NTDs) and it affects low-visibility and low-income individuals and populations in developing countries, mainly in tropical and subtropical zones [3]. Furthermore, its status as a NTD means that controlling or eradicating it by applying public health strategies remains possible, even though it has been neglected by laboratories, pharmaceutical companies and local governments.

Even 115 years after the little girl Berenice and the publication [4,5,6,7] of Carlos Chagas' work, in which he describes the etiological agent, some domestic and wild reservoirs, the vector, the life cycle, clinical manifestations and complications of the disease, CD continues to be endemic in Latin American countries, especially in Brazil, Argentina, Mexico, Bolivia and Chile [1,3]. Although the prevalence rate per 100,000 population decreased in many countries between 1990 and 2019, in Brazil an increase of 12.8% in the estimated number of cases was observed over this period [8]. The high numbers of disability-adjusted life-years (DALYs), years of life lost (YLLs) and years lived with a disability (YLDs) due to CD worldwide are still challenging. Curiously, a decrease in DALYs was observed in Brazil over the same period,

although a representative magnitude for society remained, as Brazil continues to be the country with the highest estimated number of cases, and number of deaths from the disease [8,9].

Over the past few decades, enhanced vector control programs and mandatory screening for blood transfusions have decreased the burden of this disease in Latin America [3]. However, while the typical vectorial transmission route has become controlled, the other forms of acquiring the disease have represented a challenge over the last 20 years, especially vertical, oral and post-transplant. Food-borne transmission has gained great importance, as it can cause outbreaks secondary to contamination of food and beverages and because of its pathophysiology and clinical features [10]. The vector most frequently associated with this route is *Panstrongylus geniculatus* (Latrelle, 1811) (Hemiptera, Triatominae), and transmission through this route occurs when whole triatomines or their feces carrying metacyclic trypomastigotes of *T. cruzi* taint juices made from sugar cane, bacaba, açaí or palm wine. Transmission through ingestion of infected wild-animal meats also occurs [1,2,10,11].

Currently, oral infection with *T. cruzi* is the most important transmission route in Brazil. Several outbreaks have especially been documented in the Amazon region (states of Pará and Amazonas), and in some states in the northeastern and southern regions. Furthermore, the trend over the years has been increasing [12]. This oral transmission represents a major challenge for implementation of resolution CD49.R19, created by the 49th Directing Council of PAHO in 2009, which promotes the elimination or reduction of neglected diseases and other related poverty diseases, including Chagas disease in all its forms of presentation [13].

Regarding diagnosis, there are still major challenges ahead. Some tools standardized through international guidelines are difficult to use and require several samples, and many of them are archaic. For chronic Chagas disease (CCD), use of two different serological tests continues to be recommended (enzyme-linked immunosorbent assay–ELISA, hemagglutination inhibition assay–HAI, or indirect immunofluorescence–IIF). In limited-resource settings or difficult-to-access areas, a single ELISA test may be considered, but requires confirmation with another test prior to starting treatment [14]. Over recent years, some studies have presented encouraging results regarding the performance of rapid tests (immunochromatographic) for diagnosing CCD [15,16]. However, for acute Chagas disease (ACD), we continue to rely on microscopy (with variable sensitivity) and conventional immunological tests, both often requiring several samples. To date, only one molecular test kit has been approved for use in Brazil, the NAT Chagas Kit, developed by the Oswaldo Cruz Foundation (FIOCRUZ) in partnership with the Institute of Molecular Biology of Paraná (IBMP), and its use is still far from being universal [17], and these techniques are restricted to surveillance laboratories and research centers [18]. There is also currently no point-of-care test for diagnosing ACD.

The treatment options for Chagas disease are even more restricted. The two trypanocide drugs available, benznidazole and nifurtimox, have been used for over 50 years. However, due to the long-term treatment (60 to 90 days) and high frequency of side effects (about 40%), there is a high dropout rate [19,20]. Some initiatives in this field deserve to be highlighted, such as Drugs for Neglected Disease (DNDi), which helped to develop the first formulation of benznidazole for pediatrics, in addition to promoting studies with lower doses and shorter treatment time [21,22]. However, many more actions are necessary in order to achieve the greater objective: a highly effective medication, with low cost, shorter treatment time and fewer side effects.

The present study evaluates the clinical-epidemiological characteristics and diagnostic aspects of an autochthonous outbreak of ACD due to oral transmission that occurred in the state of Maranhão, Brazil, which had previously not been considered an area of high prevalence or high risk of disease transmission. We have described the epidemiological surveillance and diagnostic methods, including molecular tools, and discussed the genetic analysis of *T*.

*cruzi*. These data have allowed us to better understand the risks associated with transmission, as well as the diagnostic and therapeutic challenges involved in the standard treatment available in Brazil.

## 2. Methods

### 2.1. Ethical statements

Formal written consent was obtained from al the patients and the parent/guardian in case of children. Ethics approval for this study was obtained from the University Hospital Research Ethics Committee of the Federal University of Maranhão, through protocol number 4.357.642, through submission through the 'Plataforma Brasil'.

This was a prospective cohort study in which patients with ACD from an outbreak due to oral transmission were followed up for 4.5 years.

### 2.2. Location

The outbreak occurred in Boa Fé village, a quilombo community, which is a type of community originally founded by fugitive slaves of African origin, located in the municipality of Pedro do Rosário (latitude -2.97727, longitude -45.3464, 2˚58'38" south, 45˚20'47" west), state of Maranhão, northeastern Brazil, in the pre-Amazon region, with a tropical climate.

### 2.3. Population evaluated and period

The study involved 39 patients with ACD from the same outbreak. The information relates to the period during which the outbreak occurred (2018) and another period 4.5 years later (2022). It was not possible other time point analysis due to the municipalities and state resource limitations.

### 2.4. Inclusion and exclusion criteria

The inclusion criteria were acceptance to participate in the study, signing of the consent form, and at least one of the following: presence of symptoms; positive direct microscopic examination of peripheral blood with *T. cruzi*; seroconversion of IgG; two times increase in IgG titer with at least 15 days interval. The exclusion criterion was refusal to participate in the study.

### 2.5. Clinical evaluation

The following were investigated: incubation period, time taken to initiate the therapy, duration of therapy and clinical findings (fever, asthenia, chills, myalgia, facial edema, dyspnea, headache, cough, abdominal pain, lower limbs edema, thoracic pain, diarrhea and inoculation chagoma).

### 2.6. Diagnostic criteria and tools

To define ACD cases, diagnostic criteria already established in the literature and guidelines were used [15]: (a) laboratory criteria (at least one of the following tests: positive direct microscopic examination of peripheral blood smear, seroconversion of IgG or 2x increase in IgG titer with at least 15 days interval); (b) clinical-epidemiological criteria (clinical findings, epidemiological exposure and at least one positive IgG finding). The direct microscopic examination consisted of use of the technique of blood smears from peripheral blood with Giemsa staining from Merck; and the serological tests used were ELISA from Bioclin, IIF from Bio Manguinhos and HAI from Wama Diagnostica. It was not possible to perform the IgM test

due to lack of resources and the inherent difficulty in the technique. Additionally, *in house* conventional PCR was performed on 33 of the 39 stored samples. A full description of the PCR (S1 Text) including electrophoresis (S1 Fig), the phylogenetic tree (S2 Fig) and the blood smear techniques (S2 Text) is presented.

### 2.7. Epidemiological surveillance

An investigative process was conducted, consisting of interviews among the local population, observation of the production process of the drink involved in the outbreak and distribution of triatomine information stations (Portuguese acronym PIT) for six months. PITs are epidemiological surveillance strategies to monitor vectors. They consist of boxes containing visual information about triatomine species that are distributed across the localities of concern. These boxes are disseminated among the population. Whenever a resident finds a suspicious arthropod, they deposit it in a PIT and report this finding to surveillance workers, who, in turn, analyze the insect and look for *T. cruzi* in its feces.

### 2.8. Treatment

All 39 patients were treated with benznidazole: adults with 5 mg/kg/day (maximum of 300 mg/day) and children with 5–10 mg/kg/day, for at least 30 days.

### 2.9. Long term follow-up

After an interval of 4.5 years, serological tests (ELISA and IIF) were performed on 26 patients. The others were lost to follow-up because they had changed their place of residence to another state. It was not possible other time point analysis due to the municipality's resource limitations.

Fig 1 show the study selection and follow-up.

## 3. Results

### 3.1. Outbreak report

On February 15, 2018, a 69-year-old woman from Boa Fé village, in Pedro do Rosário, Maranhão, Brazil, who had previously been admitted to and discharged from a municipal hospital twice, with two negative malaria blood smears, came to the Regional Epidemiological Surveillance Center of Pinheiro, Maranhão, Brazil, to undergo a third malaria blood smear. The patient had an 11-day history of abdominal pain, myalgia, asthenia, chills and fever in the late afternoon. The first two tests, previously found to be negative for malaria, were reviewed. Both of them tested positive for trypomastigote forms of *T. cruzi*, thus confirming that the patient had ACD (Fig 2). There was no need to perform the third sample. She was treated with 300 mg/day of benznidazole and became asymptomatic in less than a week.

An epidemiological investigation found that other people living in the same village also had similar symptoms. Moreover, they all had attended a celebration on January 21 (three weeks earlier) at which bacaba juice was served, which was the likely transmission route. Within two weeks after the first case, 85 people were evaluated looking for symptoms and performing serology and direct parasitological examination. An outbreak of 39 cases of ACD was diagnosed and treatment was started for all these individuals.

### 3.2. Epidemiological investigation

An investigation team was deployed to the area. Boa Fé village is a quilombo community, which is a type of community originally founded by fugitive slaves of African origin, located in

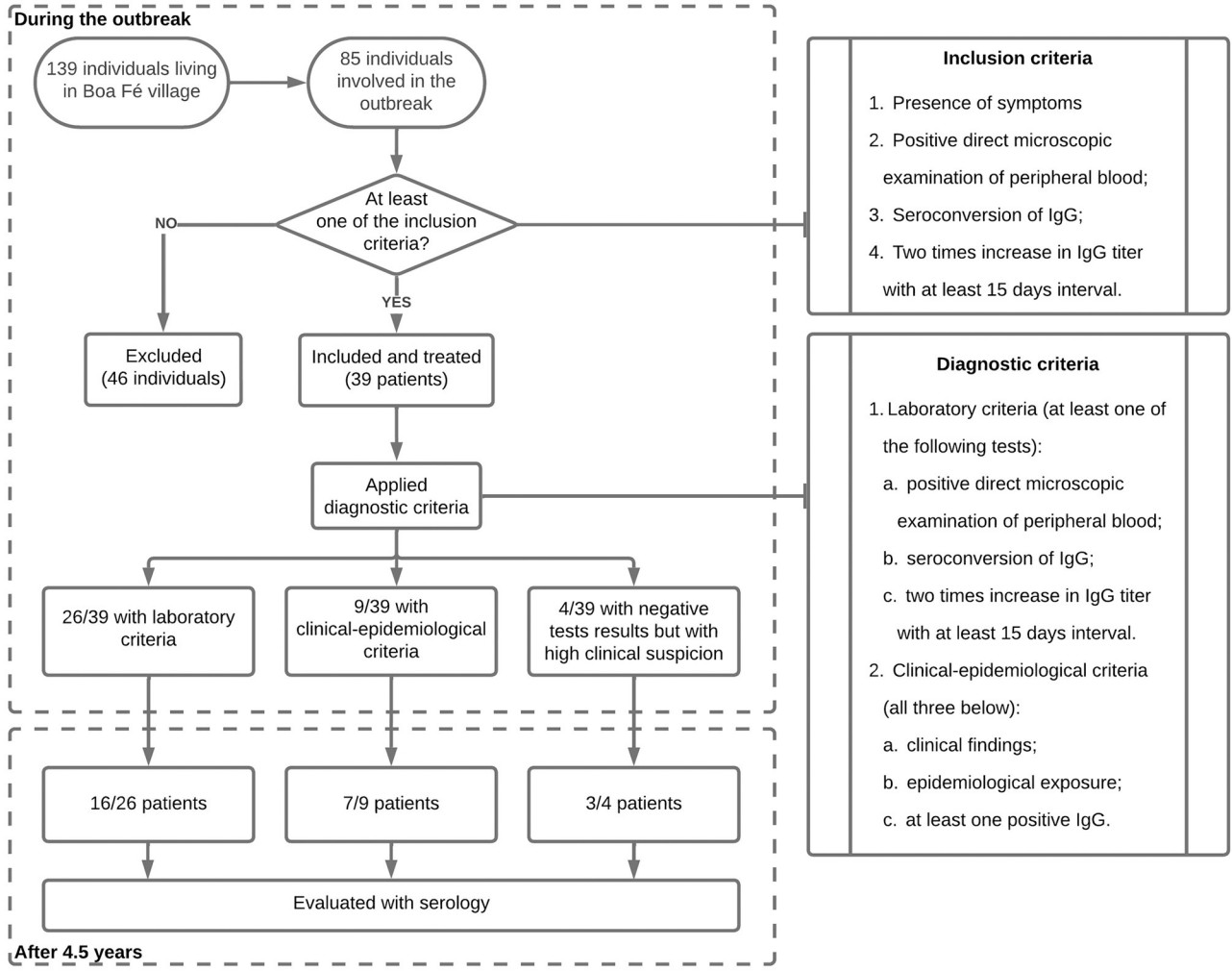

**Fig 1. Study selection, diagnostic criteria, and follow-up of the population affected by the outbreak of acute Chagas disease in Pedro do Rosário, Maranhão, Brazil.**

the municipality of Pedro do Rosário (latitude -2.97727, longitude -45.3464, 2˚58'38" south, 45˚20'47" west), which is 178 kilometers from the capital São Luís, state of Maranhão, north-eastern Brazil (Fig 3), in the pre-Amazon region, with a tropical climate. There was no paved road access. It is close to an environmental protection area (EPA), the Baixada Maranhense EPA, which is a Ramsar site.

The residents of this village live in extreme poverty, in mud houses (Fig 4) with no basic sanitation (no sewage disposal system and no treated water) and with few healthcare resources. The area is inhabited by several species that act as reservoirs of *T. cruzi*, but the one most likely involved in this outbreak was the "mucura", a marsupial species of opossum from the Didelphidae family [23], which has the habit of living or hiding in the palm tree canopy. Triatomines, which feed on the blood of these mammals, also make their home in the palm canopy. When the fruit is extracted and processed, the insects may be accidentally crushed and those infected by *T. cruzi* end up contaminating the beverage. The most frequently species in the state are from genera *Pangystrogylus* and *Rhodnius* [24]. However, the data is old, as routine monitoring is not being performed by the municipalities and the state.

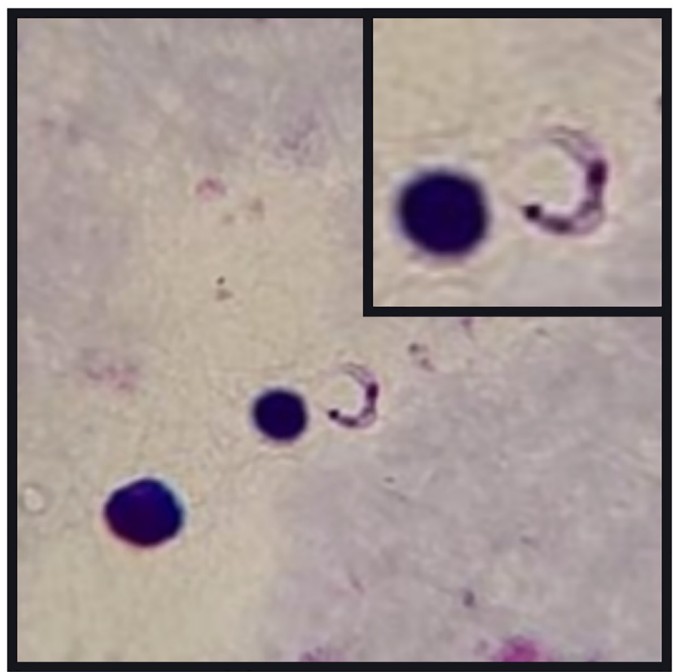

**Fig 2. Trypomastigote form of *T. cruzi* (arrow) in a peripheral blood smear stained using the Giemsa stain technique, from the index case (optical microscopy, 1000x magnification).**

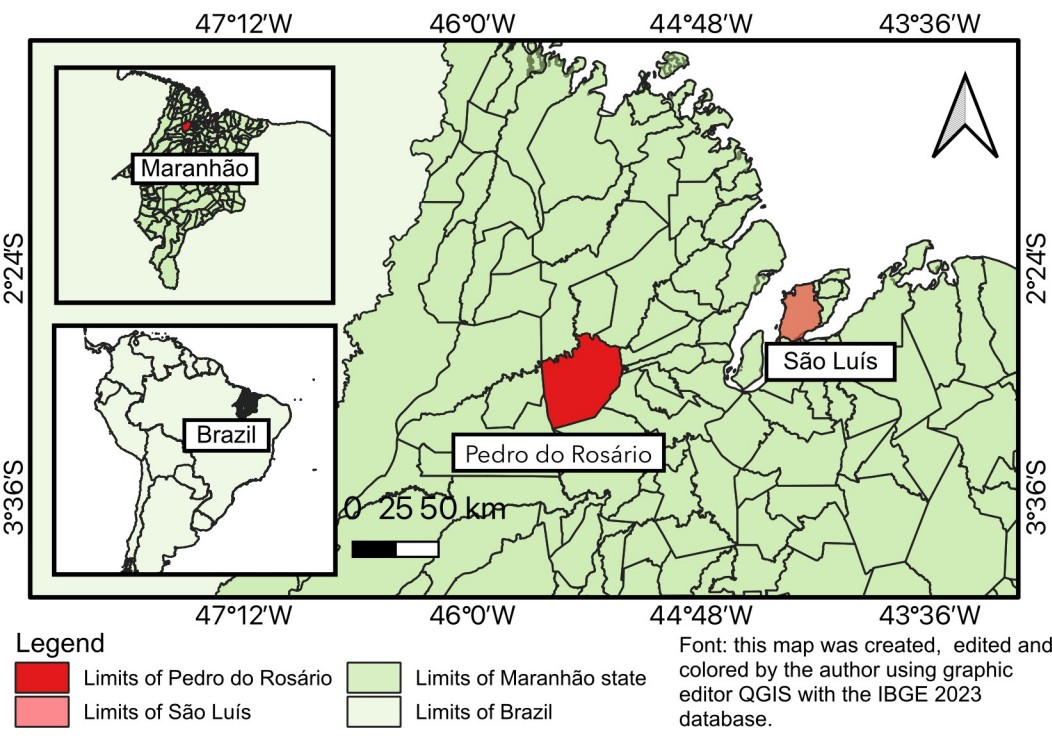

Legend

■ Limits of Pedro do Rosário  ■ Limits of Maranhão state

■ Limits of São Luís  ■ Limits of Brazil

Font: this map was created, edited and colored by the author using graphic editor QGIS with the IBGE 2023 database.

**Fig 3. Map of the geographic location of Pedro do Rosário, state of Maranhão, northeastern Brazil.** Base layer of the map (open data source): https://portaldemapas.ibge.gov.br/portal.php#homepage. Terms of use of IBGE data: https://biblioteca.ibge.gov.br/visualizacao/livros/liv101675.pdf.

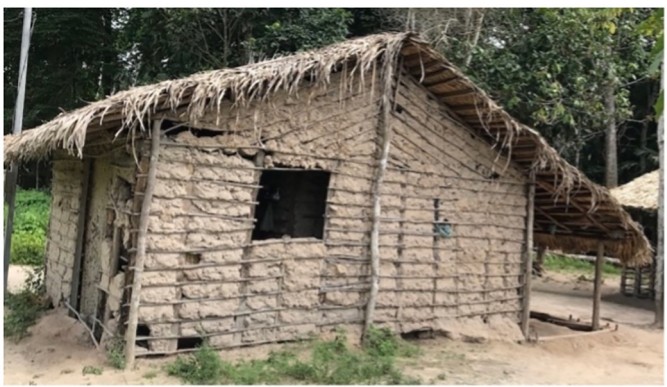

**Fig 4. Typical house in the Boa Fé village (made of mud).**

*Oenocarpus bacaba* ("bacaba") is a palm tree native to the Amazon and its fruit resembles that of *Euterpe edulis* ("juçara" or "açaí"). It can reach up to 20 meters in height and 20 to 25 centimeters in diameter (Fig 5). Its fruit is widely consumed in the region and is often the only source of food for the local population. It is a tradition for this community to hold a celebration every year during the bacaba seasonal period, which occurs between December and April.

The juice production process is completely artisanal and does not follow good practices. The bacaba bunch is placed in a plastic bag on the ground where the fruits are removed. The fruits are then placed in a bucket of untreated water where it remains for a few minutes to remove excess straw and wood. The water is then removed, and the fruits are crushed by hand to extract the juice. The liquid is then passed through a sieve to remove the seeds and skins of the fruit. Finally, the liquid is slightly warmed over a wood fire, but not at high temperature. After cooling, the juice is served.

For the entomological investigation, triatomine information stations (Portuguese acronym PIT) were distributed throughout the community and kept there for six months. Through their use, 25 specimens of *Rhodnius pictipes* (Stal, 1872) (Hemiptera, Reduviidae) and two of *R. robustus* (Larrousse, 1927) were found, all in peridomestic areas, with an infection rate of 33.3% (Table 1). No specimens of any species were found inside the houses.

### 3.3. Demographic, clinical and laboratory characteristics

The demographic and clinical characteristics of the outbreak are summarized in Table 2. The patients' mean age was 33.6 years (ranging from 10 months to 78 years). There was no difference in gender distribution (20 women and 19 men). The mean incubation period was 13.8 days (ranging from 2 to 28 days). The mean length of time between symptom onset and treatment was 16.6 days (ranging from 6 to 54 days). The most frequent symptoms were fever and lymphadenopathy (90%; 35/39), asthenia (56%; 22/39), chills (51%; 20/39), myalgia (48%; 18/39), facial edema and dyspnea (38%; 15/39) and headache and cough (31%; 12/39). There were no patients with Romaña's sign.

Out of the 39 cases, 66.6% (26/39) were diagnosed through laboratory criteria and 23% (9/39) through clinical-epidemiological criteria, while 10.2% (4/39) had negative tests but were considered to be cases of ACD due to high clinical suspicion, as show in Fig 1. We performed one peripheral blood smear, and also two sequential serological evaluations 50 days apart. The positivity rates of the tests were as follows: peripheral blood smear in 69.7% (23/33) of the cases; first serological sample in 66.6% (26/39) of the cases, and seroconversion OR 2x increase

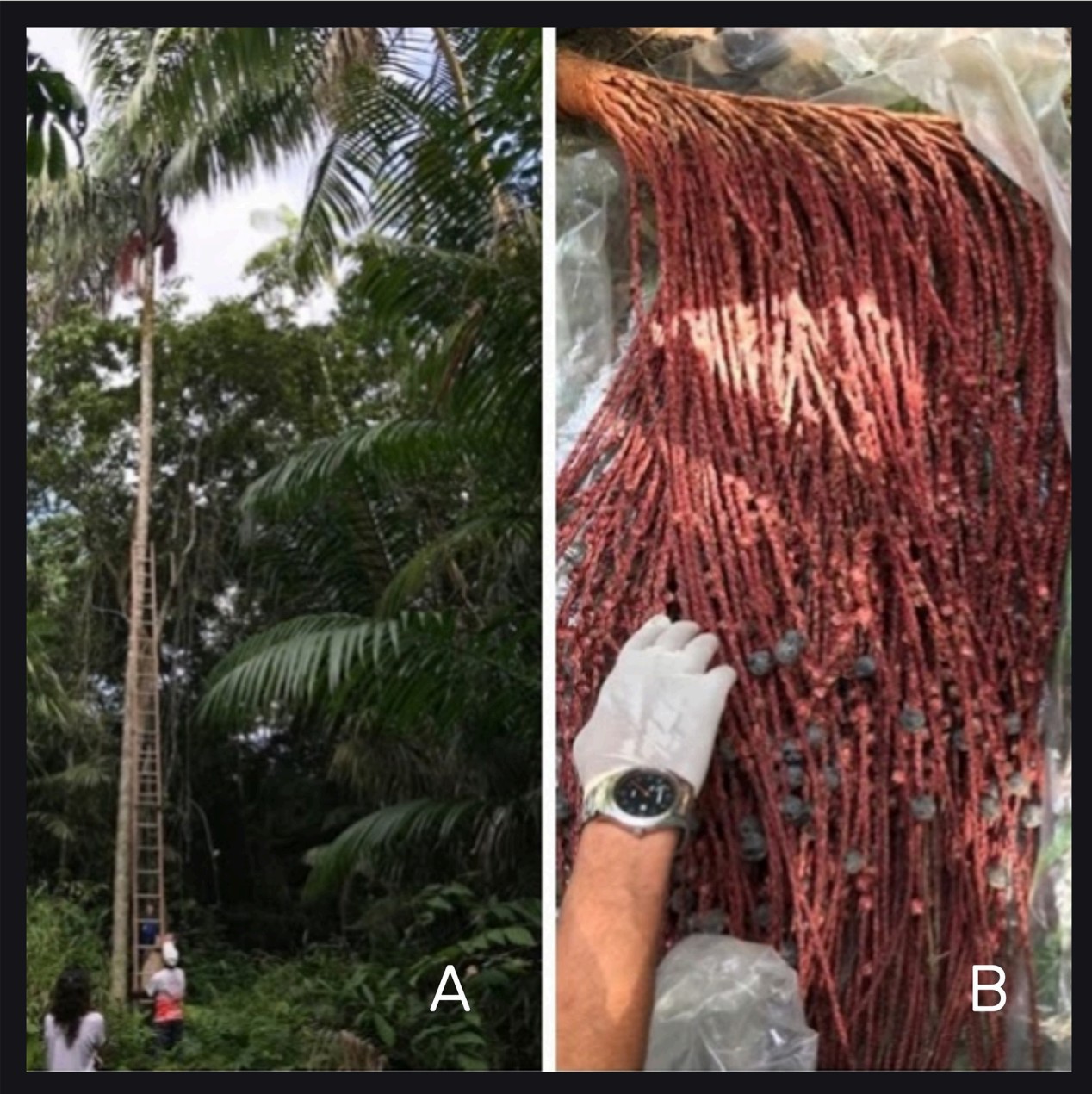

**Fig 5.** *Oenocarpus bacaba* **palm tree (A) and fruit (B).**

in IgG titer in 17.7% (6/35) of the cases. The overall positivity of the serological tests was 91.4% (32/35). Sequential samples were not collected on time in the cases of four patients.

All 33 samples (100%) amplified through PCR and sequencing with the specific primers allowed us to identify *T. cruzi*. These included nine of the thirteen patients diagnosed through clinical-epidemiological criteria and all four patients for whom all other test results were negative. The sequences from these samples have been deposited in GenBank under the following accession numbers: ON037202 to ON037234. Alignment of these sequences with *T. cruzi* sequences previously deposited in GenBank showed a genetic similarity ranging from 94.8 to

**Table 1. Quantities of triatomines captured and *T. cruzi* infection rate between March and October 2018, Pedro do Rosário, Maranhão, Brazil.**

| Months | Triatomines captured (n) | Triatomines positive for *T. cruzi* (n / %) |
|---|---|---|
| March | 3 | 0 / 0 |
| April | 5 | 1 / 20% |
| May | 3 | 1 / 33.3% |
| June | 2 | 2 / 100% |
| September | 6 | 2 / 33.3% |
| October | 8 | 3 / 37,5% |
| Total | 27 | 9 / 33.3% |

100.0% with samples from the state of Pará, which is located in the northern region of Brazil. Through phylogenetic analysis it was possible to ascertain the evolutionary relationships between our sequences and those from GenBank, as detailed in S2 Fig.

All 39 patients were treated with benznidazole: adults with 5 mg/kg/day (maximum of 300 mg/day) and children with 5–10 mg/kg/day, for 60 days. Only two patients withdrew from the treatment before the end of the schedule, but they completed at least 30 days of treatment. One of them had to be hospitalized due to a drug reaction consisting of eosinophilia and systemic symptoms (DRESS) syndrome, which was resolved with corticosteroids and antihistamines. There were no deaths due to this outbreak. The demographic, clinical characteristics, diagnostic criteria, and test results of each patient of the outbreak are summarized in S1 Table.

After an interval of 4.5 years, ELISA and IIF IgG were performed on 26 individuals, which showed that serological cure (negative tests) had been achieved in 34.6% of the cases (9/26), while decreased IgG titers were found in 15.3% of the cases (4/26). For 26.9% of cases (7/26) there was an increase in IgG titers and 15.3% of cases (4/26) maintained the same level. In 7.6% of cases (2/26) there were seroconversion. The other patients were lost to follow-up because they had changed their place of residence to another state.

**Table 2. Demographic and clinical characteristic of an outbreak of 39 cases of acute Chagas Disease due to oral transmission in Pedro do Rosário, Maranhão, Brazil.**

| Variable | Result |
|---|---|
| Gender (M:F) | 20:19 |
| Mean age (y) | 33.6 (10m-78y) |
| Mean incubation period (d) | 13.8 (2–28) |
| Mean length of time between symptom onset and treatment (d) | 16.6 (6–54) |
| Most frequent symptons (%) | fever and lymphadenopathy (90%; 35); asthenia (56%; 22); chills (51%; 20); myalgia (48%; 18); facial edema (38%; 15); dyspnea (38%; 15); headache (31%; 12); cough (31%; 12); abdominal pain (28%; 11); thoracic pain (8%, 3); diarreia (8%; 3). |

M: male; F: female; y: years; m: months; d: days; %: percentual

After this outbreak, a series of training sessions were conducted among the municipality's healthcare team and communities, with the aim of raising awareness of the importance of Chagas disease and training in safe processing, in accordance with national guides [25]. Given that the foods involved in its transmission are the only source of survival for most people in the region, stopping eating them is not an option. The population was trained to produce this food in a safer way, while maintaining the artisanal process, but this did not constitute a guidance policy extended to the entire state. The description of this protocol is included in S3 Text.

## 4. Discussion

Here, we present an epidemic outbreak of acute Chagas disease duo to oral transmission in a high-vulnerability population in the state of Maranhão, Brazil. It was the largest reported outbreak to date in this state.

In 2021, the Human Development Index (HDI) of the state of Maranhão was 0.676, the lowest in Brazil. Hunger, malnutrition, social exclusion, and deficient basic sanitation are a hallmark of the state. More than 50% of the population lives on a per capita income of less than US$81 per month. Only 56.5% of the population of Maranhão has access to a piped water system and 43.5% do not have access to drinkable water [26].

All these associated factors have given rise to an explosion of endemic diseases in this state, including increased numbers of cases and outbreaks of ACD. Until 2010, cases in Maranhão were recorded only through the classic vectorial route in a predominantly wild environment [12]. Most cases of orally transmitted ACD were imported from the state of Pará. Over the last 15 years an increase in autochthonous cases through this route has been reported, making ACD an emerging disease [12]. Cutrim (2017) reported that there was a significant increase in orally transmitted ACD from 2010 onwards [27]. The outbreak described here was the largest in the state, which emphasized that the actions of the public healthcare system must focus on promoting improvements in basic sanitation, in addition to developing a safe food production program for the population. At the time of the present study, there was no record of public policies that included safe production of these products by the population.

Orally transmitted Chagas disease has particularities that are worth mentioning. In contrast to vector-borne transmission, the higher parasite load after oral ingestion makes controlling the disease more difficult for the immune system [28]. This is demonstrated through higher parasitemia and increased tissue invasion [2]. It may lead to a shorter incubation period, [29,30], early cardiac involvement and increased severity [31,32,33]. However, the clinical manifestations may be nonspecific, which makes diagnosis a real challenge for clinicians, given that ACD can be confused with other acute febrile illnesses, such as leptospirosis, visceral leishmaniasis, malaria, enteric fever, rickettsiosis, and some arboviruses [10].

The possibility of orally acquired CD should always be suspected in situations in which an epidemiological link exists, such as in endemic areas or with more than one simultaneous case with ingestion of suspected food. A good clinical clue is the presence of fever for more than a week, associated with facial edema or bipalpebral edema and lymphadenopathy in patients with an epidemiological link. High frequency of fever and lymphadenopathy (90%) and low frequency of facial edema (38%) were found in the present study, which is in line with other reports [34,35,36].

The difficulty in establishing the diagnosis in real life was demonstrated here. The tools available that are recommended in the guidelines are low in efficiency, as they require several visits and involve complex and often rudimentary processes. The gold standard in diagnosing ACD is direct parasitological methods, but these have variable sensitivity, as they are operator-

dependent and need to be repeated on multiple samples [14, 37]. Therefore, the best recommendation is to combine methods [14].

In infectious disease, identification of the agent is often the gold standard. In Chagas disease, there are direct and indirect methods. Xenodiagnosis and blood culture are among the indirect methods. Xenodiagnosis is an old but sophisticated procedure. It was the first method used when CD was discovered. Laboratory-reared vectors are induced to bite patients, and 30 to 60 days later, their excreta are analyzed to search for the parasite. There are clearly some disadvantages in this technique, such as the time and resources required to perform the procedure, as well as the risks associated with handling insect vectors. Hemoculture is another indirect method, in which the examination should be performed monthly for as long as six months. However, both xenodiagnosis and hemoculture have low sensitivity [38,39] (around 20%) and are best suited for the chronic phase of the disease [14,37].

The most sensitive of the direct parasitological methods consists of examination of fresh blood in association with concentration methods [14,37,40] (Strout, buffy coat or microhematocrit). However, considering that the region of the present study is endemic for malaria, the blood smear technique is the tool that is most available, and it is easily performed by some laboratory technicians in Maranhão. Furthermore, it is recommended that the parasitological test should be repeated every 12–24 hours while the clinical condition persists, until the parasite is found or another diagnosis is established [14,37,40]. Unfortunately, due to the logistical difficulty in collecting samples in remote areas, it was not possible to repeat this test in a timely manner. Also, there has been a shortage of qualified technicians to perform good-quality microscopy over the years, thus posing a threat to making diagnoses of ACD. In the present study, the peripheral blood smear technique was used and positivity of 69.7% was found in a single-sample analysis, which was higher than had been found in other reports [41]. Nonetheless, variable sensitivity has been reported in the literature, with the best results found in reference laboratories [34,42,43].

In diagnosing ACD, serological tests are considered to be the tool with the greatest sensitivity and specificity, through detecting IgG seroconversion or an IgG titer that more than doubles over a minimum interval of 2 weeks [17]. However, this is not the most recommended tool, given the time required for achieving confirmation [39,40]. Moreover, use of IgM has serious limitations in ACD: there are no commercial kits; positive controls are hard to find; it remains positive for several years in chronic patients; and it might give false positives with rheumatoid factor and in cases of other infectious diseases such as leishmaniasis [44]. In the outbreak of the present report, two sequential serological tests were performed, with an interval of 50 days between them. ELISA and IIF were used for the first sample and ELISA and HAI for the subsequent sample. We found positivity of 66.6% (26/39) in the first sample and 17.7% seroconversion (6/35). Four samples were lost in sequential analysis. The overall positivity of the serological tests was 91.4% (32/35). This result was in line with those of several other publications [38,45,46]. It was not possible to perform IgM, due to the limitations mentioned above, and the reference laboratory did not have the resources to do this test. If this test had been performed, we could have found higher values, but with conflicting results.

Regarding the cure criteria, some remarks must be made. There are no gold standard serological tests or biomarkers that define situations of CD cure. Molecular biology analyses and non-conventional serological tests aimed at specific forms of *Trypanosoma* like anti-live trypomastigote (FC-ALTA), anti-fixed epimastigote (FC-AFEA) and anti-three evolutive forms (FC-ATE) or antigen tests (ELISA-F29) have been tested with inconclusive results [46,47,48,49,50]. Therefore, reversion to negative seen through conventional serological tests is considered to be the most reliable indicator of successful cure [51,52], and this is associated with the absence of parasitemia [53]. In long-term follow-up, after 54 months (4.5 years), we

were able to perform serological tests (ELISA and IIF) on 26 individuals and found serological cure (negative tests) in 34.6% (9/26) and decreased IgG titers in 15.3% (4/26). The other patients were lost to follow-up because they had changed their place of residence to another state. These rates were lower than other reports on ACD (up to 70% after 5 years) [53], considering that the total negativity or decline of antibodies achieved 4.5 years after the outbreak was only 50%. This finding brings to light the possibility of hypotheses such as pre-exposure (as some of the patients already had positive IgG in the first sample), or re-exposure to the pathogen. This could mean that patients already had the indeterminate chronic form of the disease, contributing to therapeutic failure. Moreover, if additional samples had been obtained after shorter time intervals, this would have enabled better analysis on the kinetics of antibodies. Nonetheless, it has previously been noted that the decline in IgG titers can take many years, depending on the form of infection [54,55].

Molecular diagnosis has restricted use and is performed only by research centers or reference laboratories, due to the lack of defined protocols and standardized operational procedures, as well as the scarcity of commercial kits for use in routine surveillance. To date, only one kit has been approved for use in Brazil, the NAT Chagas Kit, developed by the Oswaldo Cruz Foundation (FIOCRUZ) in partnership with the Institute of Molecular Biology of Paraná (IBMP), and its use is still far from being universal [18]. Despite the diagnostic complexity of CD in all of its phases, we obtained excellent results using conventional PCR to detect and identify *T. cruzi* in the clinical samples evaluated. These specific primers were previously described in the literature by Wincker et al. (1994) [56] and have been widely used as an adjunct to serological tests in diagnosing this disease. In a recent meta-analysis, in which several molecular biology methods were compared, Pascual-Vázquez et al. (2023) [57] demonstrated that it is not yet possible to determine the most effective molecular protocol because of the heterogeneity of the studies. In the present study, positivity was found through conventional PCR in 100% of the samples tested, even in patients whose other tests were all negative. This result is in line with reports on other outbreaks [58,59,60]. There is an urgent need, for standardization of at least one molecular biology technique, so that this method can be used for routine diagnosis of ACD.

Genotyping is a powerful tool for investigating orally transmitted outbreaks of CD. *Trypanosoma cruzi* is divided into at least seven lineages, as discrete typing units (DTUs), from TcI to TcVI and Tcbat (there is no consensus regarding the definition of the latter as a distinct DTU) [61,62,63]. Velásquez-Ortiz et al. (2022) demonstrated that DTUs are widely distributed across all countries, and in Brazil most cases are associated with TcI [64]. However, it is important to highlight that other genotypes have also been reported, like TcIII and TcIV [60,65,66]. In northeastern Brazil, the most frequent forms are TcI and TcIV [67]. The present study did not have the aim of analyzing the clonal aspect of *T. cruzi* strains. However, construction of a phylogenetic tree containing 33 sequences from this outbreak together with another 77 sequences extracted from another Brazilian outbreak from Pará state showed a genetic similarity ranging from 94.8 to 100.0%, demonstrated that the evolutionary distances are low and may represent a common source. Molecular studies are needed in order to better evaluate the transmission dynamics or origin of these strains.

In the oral transmission route, the triatomine is accidentally macerated together with fruit or food is directly contaminated by its feces when left unprotected. The species most frequently involved in Colombia and Venezuela is *P. geniculatus*. In Brazil, greater variability of species has been found, but most of them have belonged to the genera *Triatoma* and *Panstrongylus* [68]. Entomological surveillance is of great importance for better understanding the life cycle and persistence of *T. cruzi* in nature. We distributed PITs throughout the community and kept them there for six months. In this study, 27 viable triatomines were found, of which more than

90% were *Rhodnius pictipes*, all in peridomestic areas. No species were found inside houses. Previously, only two outbreaks (in Brazil and Colombia) had reported this vector as probably involved in transmission [69,70], but its ability to transmit the disease was recently demonstrated in an experimental study [71]. The infection rate was 33.3%, which despite being higher than in most surveillance in Brazil, was in line with findings from studies in other states, where rates above 30% and up to 70% were found in peridomestic areas [72,73,74].

All the patients were treated with the only drug available in Brazil, benznidazole. This is a medication with a high rate of adverse reactions. Fortunately, suspension of the medication was only necessary due to adverse reactions in the cases of two patients, and both of these individuals received the treatment for 30 days, a period considered adequate. However, management of the treatment was quite arduous, given the need to use symptomatic medication in most patients. The MULTIBENZ trial [75], a randomized, double-blind, phase 2b multicentric trial, tested benznidazole treatment in three different regiments (300 mg/day for 60 days, 150 mg/day for 60 days, and 400 mg/day for 15 days) with similar results regarding parasitological responses. This suggest that reducing dosage or duration of treatment can contribute to better adherence and greater coverage. It is imperative that new initiatives, such as DNDi, are funded to develop new drugs or shorter therapeutic regimens.

The lethality of an outbreak is quite variable and can be influenced by four factors: the phenotype of the pathogen; the ingested parasite load; the individuals' susceptibility; and whether treatment was started early on. A recent meta-analysis demonstrated that the overall lethality of ACD is around 1%, but it also showed that lethality can reach up to 30% in large outbreaks (with more than 10 patients) [76]. It was not possible to analyze the first three factors mentioned above. Regarding early treatment, the mean length of time between symptom onset and treatment was 16.6 days (ranging from 6 to 54 days). This demonstrates that action was implemented quickly by the surveillance center, which may have had an impact on the absence of lethal outcomes.

Most orally transmitted ACD cases have occurred after ingestion of fresh juices, like açaí (*Euterpe oleracea*) and sugar cane (*Saccharum spp.*). However, other beverages have also been cited [76,77]. In this event, the most probable source was bacaba juice. This was the first time that this fruit had been reported to be involved in an outbreak in the state of Maranhão but reports of this in Venezuela and another Brazilian state have already been published [66,78]. At first sight, it may seem that this is a problem restricted to remote and highly vulnerable locations, but a study demonstrated the presence of *T. cruzi* in açaí products sold in markets in Rio de Janeiro, Brazil, which is well outside the Amazon region [79]. Furthermore, from the perspective of the "One Health approach", with intense human migration and exporting of natural products from endemic areas, CD can already be considered to be a global health problem: papers published in European and North American countries have demonstrated this concern [80]. It is also important to mention a study that demonstrated the presence of infected triatomines in a densely populated area in the northeastern region of the United State, a nonendemic area [79]. Therefore, there is an urgent need to implement safe food production programs, aligned with better diagnostic and therapeutic tools.

After this outbreak, a series of training sessions were conducted among the municipality's healthcare team and communities, with the aim of raising awareness of the importance of Chagas disease and training in safe processing, in accordance with national guides [26]. Given that the foods involved in its transmission are the only source of survival for most people in the region, stopping eating them is not an option. The population was trained to produce this food in a safer way, while maintaining the artisanal process, but this did not constitute a guidance policy extended to the entire state.

The limitations of this study are all related to the resource limitations of the municipalities and the state. It was not possible to perform an adequate cardiological assessment on all patients, restricting the clinical analysis; it was not possible to carry out other collections in a shorter period of time, making it impossible to analyze the serological kinetics; nor was it possible to collect new PCR samples.

This event demonstrates that there is still a long way to go before this disease is controlled. As mentioned previously, the status of CD as a neglected tropical disease means that controlling or eradicating it through applying public health strategies remains possible. Despite the contemporary nature of the difficulties described here, they remain very similar to those at the time of its discovery. Even after more than a century, Carlos Chagas is still right when he said:

"Will it be possible, within public hygiene, to find efficacious ways of attenuating this affliction? We believe so, if this problem—most surely a problem of the State and of humanity—becomes the concern of a scientifically well-advised statesman."

(Chagas, 1917) [81].

As globalization breaks all borders through the migration of people and the exporting of raw materials, we postulate here that we must spread Carlos Chagas' thoughts throughout the world.

## 5. Conclusions

Chagas disease is a classic NTD with a special characteristic: it is called "the silent disease". The ability of this disease to keep the host virtually asymptomatic for a long time allows it to thrive. In this study, we present all the barriers encountered in managing real-life situations of an acute event: the vulnerability of the population; the dependence on old diagnostic techniques, together with little interest in developing microscopy professionals; lack of standardization of molecular biology techniques; few therapeutic options, with many side effects; and the neglect of governments to perform the basic testing and adequate follow up of affected people. The latter ended up being the limiting factor of this study, as it was not possible to perform basic cardiological assessment tests in all patients, serial serologies in shorter times and other molecular biology samples. We demonstrated that rapid surveillance action with early initiation of treatment culminated in an outbreak without deaths. To improve early treatment, it is necessary to invest in better diagnostic techniques and make them widely available. We also demonstrated and propose that polymerase chain reaction should be standardized within the diagnostic routine in all cases with negative parasitology. Acute Chagas disease is a threat to public health. While advances in vector control have been achieved, the peridomestic and sylvatic cycle remains active, thus perpetuating the parasite in nature. The most vulnerable populations continue to be those in extreme poverty, without access to basic sanitation, or who live near the Amazon rainforest. However, as the beverages that are classically involved in oral transmission are increasingly exported to other regions, and with the intense human migration that has been occurring increasingly, outbreaks in non-endemic regions are an additional concern in relation to controlling CD. Until everyone looks at the problem from a "One Health" perspective, no nation is safe from Chagas disease.

## Supporting information

**S1 Fig. Electrophoretic profiles of PCR products of the variable region of the kDNA mini-circle obtained with DNA from some clinical samples of patients with Chagas disease.** (DOCX)

**S2 Fig. Phylogenetic tree of the 33 samples from the outbreak along with 77 samples from another outbreak.**
(DOCX)

**S1 Text. Methodology of polymerase chain reaction.**
(DOCX)

**S2 Text. Methodology of blood smear technique.**
(DOCX)

**S3 Text. Methodology of safe preparation of bacaba juice.**
(DOCX)

**S1 Table. Demographic, clinical and laboratory characteristics of 39 cases of acute Chagas disease.**
(DOCX)

## Acknowledgments

The authors thank the State Health Department of Maranhão (SES/MA) and the Oswaldo Cruz Institute—Central Public Health Laboratory of Maranhão (IOC-LACEN/MA) for their logistical support and analyses on laboratory tests. The authors also thank the Foundation for Support of Scientific Research and Technological Development of Maranhão (FAPEMA) and the National Council for Scientific and Technological Development (CNPq).

## Author Contributions

**Conceptualization:** Eudes Alves Simões-Neto, Jackson Maurício Lopes Costa, Conceição de Maria Pedrozo e Silva de Azevedo.

**Data curation:** Eudes Alves Simões-Neto.

**Formal analysis:** Eudes Alves Simões-Neto, Conceição de Maria Pedrozo e Silva de Azevedo.

**Investigation:** Eudes Alves Simões-Neto, Maria Rosa Quaresma Bomfim, Jackson Maurício Lopes Costa, Amanda Ferreira Simões, Domingos Carvalho Sodré, Ana Cleide Mineu Costa, Samuel Vieira Rodrigues Dumont, Bruna de Oliveira de Melo.

**Methodology:** Eudes Alves Simões-Neto, Daniel Wagner de Castro Lima Santos, Maria Rosa Quaresma Bomfim.

**Project administration:** Eudes Alves Simões-Neto, Conceição de Maria Pedrozo e Silva de Azevedo.

**Resources:** Maria Rosa Quaresma Bomfim.

**Supervision:** Conceição de Maria Pedrozo e Silva de Azevedo.

**Validation:** Daniel Wagner de Castro Lima Santos, Maria Rosa Quaresma Bomfim, Conceição de Maria Pedrozo e Silva de Azevedo.

**Visualization:** Eudes Alves Simões-Neto, Amanda Ferreira Simões.

**Writing – original draft:** Eudes Alves Simões-Neto, Lucas Dias Vasconcelos.

**Writing – review & editing:** Eudes Alves Simões-Neto, Daniel Wagner de Castro Lima Santos, Maria Rosa Quaresma Bomfim, Conceição de Maria Pedrozo e Silva de Azevedo.

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
