## [Decision Letter · Decision Letter 0]

17 Jun 2024

Dear Prof Alves Simoes Neto,

Thank you very much for submitting your manuscript "An outbreak of Chagas disease through oral transmission: 115 years after discovery by Carlos Chagas, the disease is still hard to manage." for consideration at PLOS Neglected Tropical Diseases. As with all papers reviewed by the journal, your manuscript was reviewed by members of the editorial board and by several independent reviewers. The reviewers appreciated the attention to an important topic. Based on the reviews, we are likely to accept this manuscript for publication, providing that you modify the manuscript according to the review recommendations. 

Sincerely,

Andrés F. Henao-Martínez, M.D.

Academic Editor

Susan Madison-Antenucci

Section Editor

Reviewer's Responses to Questions

**Key Review Criteria Required for Acceptance?**

**Methods**

-Are the objectives of the study clearly articulated with a clear testable hypothesis stated?

-Is the study design appropriate to address the stated objectives?

-Is the population clearly described and appropriate for the hypothesis being tested?

-Is the sample size sufficient to ensure adequate power to address the hypothesis being tested?

-Were correct statistical analysis used to support conclusions?

-Are there concerns about ethical or regulatory requirements being met?

Reviewer #1: The methods used to develop the study were adequate.

Reviewer #2: PITs are a strategy used by the Oswaldo Cruz Foundation (FIOCRUZ) to monitor vectors.

PITs are epidemiological surveillance strategies. And not from FIOCRUZ.

Reviewer #3: -Are the objectives of the study clearly articulated with a clear testable hypothesis stated? YES

-Is the study design appropriate to address the stated objectives? YES

-Is the population clearly described and appropriate for the hypothesis being tested? YES

-Is the sample size sufficient to ensure adequate power to address the hypothesis being tested? YES

-Were correct statistical analysis used to support conclusions? YES

-Are there concerns about ethical or regulatory requirements being met? NO

Reviewer #4: Overall, the study's objectives are clearly articulated, providing a well-defined and testable hypothesis. However, it would be appropriate for the authors to explain why they considered analyzing only two time points: before the treatment and 4.5 years after the treatment. Why did they not consider other time points after the treatment? (lines 167-168)

In general, the study population is briefly described, but in another section of the manuscript. The authors should conduct an extensive review of the text, as information that should be in the methodology section is found in other sections. For example, the data contained in lines 221-222 (related to the results section) should be in the methodology. It would also be useful to include a table in the methodology section that gathers data on the number of patients, their age, gender, the time taken to receive treatment, and the therapeutic regimen they underwent. This information is present in the text but in a different section and not very detailed, which makes the presentation of the data quite confusing.

It is important for the methodology to be more thoroughly explored. Information about the outbreak is neither in the introduction nor the methodology but is presented throughout the results. The authors should review the manuscript text, which needs to be restructured. Additionally, we need more data about the outbreak: where does it occur? Provide a map. How many people are involved? Provide the latitude and longitude of the location. Describe the characteristics of the location, etc. Some of this information is in the results section and should be in the methodology, while other information is missing from the manuscript and should be included.

In line 174, the authors mention criteria established in the literature and guidelines but do not specify which ones. Please cite them. Suggestion: present the criteria (lines 174-178) in a table.

The authors describe the cPCR in the Supporting Information section.

Reviewer #5: The objectives are clear and practical, the study population is delimited and the size of the population does not affect the results. Ethics protocols could be better clarified, such as the use or not of the 'Plataforma Brasil'.

In 'supporting information' all protocols need details and not just the PCR.

Reviewer #6: yes to all the above. No concerns about ethics.

**Results**

-Does the analysis presented match the analysis plan?

-Are the results clearly and completely presented?

-Are the figures (Tables, Images) of sufficient quality for clarity?

Reviewer #1: The resulsts are well presented.

Reviewer #2: -After an interval of 4.5 years, ELISA and IIF IgG were performed on 26 individuals, which showed that serological cure had been achieved (negative tests) in 34.6% of the cases (9/26), while decreased IgG titers were found in 15.3% of the cases (4/26).

What about the others evaluated?

- After this outbreak, a series of training sessions were conducted among the municipality's (...)

I suggest this part is in the results section

Reviewer #3: -Does the analysis presented match the analysis plan? YES

-Are the results clearly and completely presented? YES

-Are the figures (Tables, Images) of sufficient quality for clarity? YES

Reviewer #4: Since there are missing details in the methodology, the results are consequently presented in a confusing manner. For example, subsection 3.1 (lines 204-222) should be rewritten in the methodology. It is also important to include how the 85 individuals investigated due to the outbreak were assessed (line 221).

Additionally, subsection 3.2 (lines 224-267) should also be in the methodology, and more information should be added, such as: what is the climate type of the location? What type of vegetation is there?

Line 235: The authors should add an image of the 'mucura.'

Lines 237-238: What are the predominant species of triatomines in the study area?

Line 247: What is the seasonal period of bacaba? This should be mentioned in the text.

Lines 259-260: How was the infection analysis of the triatomines conducted?

Lines 268-275: Suggestion: a graph could be plotted showing the frequencies of the symptoms.

Table 2 (line 277) is too extensive, making it difficult to understand the results and compare the different diagnoses. Suggestion: the table could be split to separately present the different criteria and the results before treatment and 4.5 years after treatment.

The first information about the sequencing was only presented in the results (line 289). There should be information about the sequencing in the methodology. Why and how was the sequencing performed? Were they able to identify the different DTUs of T. cruzi?

Reviewer #5: The results section is complete and adequately presents the same

Reviewer #6: Figure 1 could be specifically cropped to show the trypomatigote more clearly.

**Conclusions**

-Are the conclusions supported by the data presented?

-Are the limitations of analysis clearly described?

-Do the authors discuss how these data can be helpful to advance our understanding of the topic under study?

-Is public health relevance addressed?

Reviewer #1: The work is relevant and the conclusions were supported by the data presented. Limitations of the analysis were not described.

Reviewer #2: (No Response)

Reviewer #3: -Are the conclusions supported by the data presented? YES

-Are the limitations of analysis clearly described? YES

-Do the authors discuss how these data can be helpful to advance our understanding of the topic under study? YES

-Is public health relevance addressed? YES

Reviewer #4: In the conclusion of the manuscript, the authors call for a reflection on Chagas disease, which is highly important, especially with the increase in cases and outbreaks observed in the region and in other endemic and non-endemic areas. However, the authors should discuss more about the methodologies proposed for the molecular diagnosis of Chagas disease, considering they propose the methodology used in the study, which is conventional PCR (described in 1989 by Wincker), as the gold standard method. Although this methodology is well established, the authors do not explore it much throughout the manuscript, nor do they discuss and compare it with other already described molecular methodologies. Reading the 2nd Brazilian Consensus on Chagas Disease, 2015, published in 2016, as well as other related articles, could enrich the discussion about the proposal made by the authors. I also recommend the reference Moreira et al., 2023 (Validation of the NAT Chagas IVD Kit for the Detection and Quantification of Trypanosoma cruzi in Blood Samples of Patients with Chagas Disease), as it deals with a commercial kit approved by ANVISA for molecular diagnosis through real-time PCR of patients with Chagas disease.

The arguments in lines 380-384 and 389-392 are repetitive and redundant. The authors should combine them into a single justification for not performing IgM analysis in these patients.

Lines 407-408: Could it not be related to therapeutic failure? The MULTIBENZ study (Bosch-Nicolau et al., 2024) explores this.

Lines 412-414: Currently, the commercial NAT Chagas kit (Moreira et al., 2023) is in circulation.

Lines 435-438: What are the most common DTUs?

Reviewer #5: I found the discussion extensive and could present a clear and objective discussion.

Reviewer #6: Conclusions are supported by the data.

Limitations of diagnostic tests are discussed.

Public health relevance cleearly highlighted.

**Editorial and Data Presentation Modifications?**

Reviewer #1: The study presents health public relevance. Some minor modifications are required:

1. Present a paragraphy describing the limitations of the study

2. The discussion is too long and repetitive. It could be reduce to 2/3 lenght

3. Add a suggestion of protocol containing PCR as a diagnosis test to ACD

Reviewer #2: (No Response)

Reviewer #3: (No Response)

Reviewer #4: Minor modifications:

Abstract: Line 46 – Correct “Ita” to “Its”

Introduction: Lines 101-103 - Lack of reference for the information provide herein. Please include.

Lines 137-139 - Why not use qPCR methodology instead of cPCR?

Line 144: It would be interesting to highlight the challenges of treatment with these medications in chronic patients.

Discussion: Lines 320-322 - Lack of reference for the information provide herein. Please include.

Line 378: Lack of reference for the specificity of serological tests. Please include.

Supplementary: Line 803: What is the size in base pairs of this product?

Line 812: Is it just boric acid or is it a buffer with Tris, boric acid, and EDTA? Also, what concentration is used? 

About the results of cPCR: Is there a representative gel for this supplementary information? Were negative controls for extraction and PCR used, as they are crucial for molecular diagnosis?

After sequencing, which DTUs were found in these patients?

Reviewer #5: Minor revision.

Reviewer #6: A few minor changes (mainly clarification) are recommended before acceptance

1. Line 85. “executability” - is practicality a better term?

2. Line 108. Please clarify what is meant by “although a representative magnitude for society remained”.

3. Line 112. Please specify which forms of transmission remain a challenge in the authors’ opinion: oral, transplacental, sexual, via blood transfusion or via organ transplantation? 

4. Lines 205 to 212. As written, it is not clear if the third malaria blood smear showed the presence of trypomastigotes and that this observation prompted a re-examination of the first two smears, which were subsequently also found to be positive for Trypanosoma cruzi.

5. Line 217. Would it be possible to enlarge and crop the area of the blood smear to show the trypomastigote while retaining a reasonable resolution in Figure 1?

6. Line 224. Not all readers will know that a quilombo is a settlement originally founded by fugitive slaves of African origin. Please expand to clarify.

7. Line 274. I presume the facial edema was not associated with Romaña’s sign? It might be useful to specifically state that it was absent (if that was the case) as this reduces the possibility of vector transmission in these cases.

8. Line 291-292 and supporting information. Comment on how cross-contamination was ruled out as a cause of a positive PCR result when all other tests were negative.

9. Lines 489 to 490. Some specific details of the recommended pasteurisation method (and references to experimental validation) might be helpful here, since this does not specifically clarify how this differs from the method on line 255 where the liquid is slightly warmed but not boiled.

10. References. The list is comprehensive and appropriate. Please ensure that all titles are in sentence case not title case and that all species names are italicised.

**Summary and General Comments**

Reviewer #1: (No Response)

Reviewer #2: The article is well written and with all the relevant information. However, I believe that some aspects could be improved.

Abstract: -Ita expansion to Other 

Please check the text. I believe there is something written wrong.

Reviewer #3: Dear authors, 

The manuscript is well structured, with appropriate language and references. The outbreak of oral infection is well documented, followed over a long period.

As strengths I highlight: 

- the importance of PCR for diagnosing acute cases of Chagas disease; 

- the importance of PITs for contributing to the sampling of triatomine species;

- long period of patient follow-up.

As weakness I highlight: the lack of a PCR test at the end of the 4.5 years of follow-up. I believe it would be interesting to observe the efficiency of PCR at a later stage of infection. 

Below are some small suggestions to contribute to the work:

Line 45: I believe that "however" is not the best conjunction considering the first sentence. As a suggestion, I believe it could be kept in the same sentence: “(...) subtropical zones, and even more (...)”

Line 46: Change “Ita” to “It’s”

Line 92: Change “Trypanosoma cruzi” to “Trypanosoma cruzi (Chagas, 1909) (Kinetoplastida, Trypanosomatidae)”

Line 115: Change: “Panstrongylus geniculatus” to “Panstrongylus geniculatus (Latreille, 1811) (Hemiptera, Triatominae)”. Please add the author and year to other scientific names that are mentioned for the first time.

Line 317: Please show the currency of the value

Line 427: At the beginning of the sentence, the species name is not abbreviated

Best regards

Reviewer #4: This paper provides important information and data about Chagas disease and the most relevant transmission methods at the moment. However, the authors should conduct an extensive review of the manuscript as it needs restructuring. Additionally, enriching the discussion with more current data is important so they can propose cPCR methodology as the gold standard.

Reviewer #5: I understand the problem and see that the manuscript is necessary.

Reviewer #6: This is a well presented and clearly written article describing the diagnosis, treatment and control of an outbreak of Chagas disease via oral transmission in northeastern Brazil. The contents of this article will be of considerable interest to the readership, particularly the extensive discussion of the limitations of currently used diagnostic techniques and test of cure through parasitological, serological and molecular genetic methods. The need for better, safer drug treatments and the need for implementation of effective food safety measures, appropriate for areas of extreme poverty are also appropriately highlighted.

PLOS authors have the option to publish the peer review history of their article (what does this mean?). If published, this will include your full peer review and any attached files.

Reviewer #1: No

Reviewer #2: Yes: Ariela Mota Ferreira

Reviewer #3: No

Reviewer #4: No

Reviewer #5: No

Reviewer #6: No

Figure Files:

Data Requirements:

Reproducibility:

References

---

## [Editor Report · Decision Letter 1]

9 Jul 2024

Dear Prof Alves Simoes Neto,

Thank you very much for submitting your manuscript "An outbreak of Chagas disease through oral transmission: 115 years after discovery by Carlos Chagas, the disease is still hard to manage." for consideration at PLOS Neglected Tropical Diseases. As with all papers reviewed by the journal, your manuscript was reviewed by members of the editorial board and by several independent reviewers. The reviewers appreciated the attention to an important topic. Based on the reviews, we are likely to accept this manuscript for publication, providing that you modify the manuscript according to the review recommendations. 

* Please refine the manuscript for clarity throughout (suggest to avoid passive voice for example, and redundancy in the sentences). Feel free to aid from softwares supported by AI or grammarly. See below an example of a rewrited abstract. 

* Please add an aim at the end of the background in the abstract.

* Please add a geography area of the outbreak and source of transmission to the abstract, and perhaps the title

* Please respond to reviewer #6, change 2. Line 108

* Suggest to reconsider adjusting the title using key words and more descriptive context for the outbreak. It's unclear the goal of mentioning the 115 years. Example: Oral Chagas disease outbreak by bacaba juice ingestion in the municipality of Pedro do Rosário, state of Maranhão, Brazil: diagnosis and treatment challenges. This can help with future citation and search engines captures. 

Sincerely,

Andrés F. Henao-Martínez, M.D.

Academic Editor

Susan Madison-Antenucci

Section Editor

Figure Files:

Data Requirements:

Reproducibility:

Abstract example:

Background: Orally transmitted acute Chagas disease (ACD) primarily affects low-visibility and low-income individuals in tropical and subtropical zones. Managing ACD remains challenging. Its spread to non-endemic areas has made it a global health issue.

Methodology and findings: This report examines an outbreak of 39 ACD cases due to oral transmission. A clinical and epidemiological investigation, including an entomological search, was conducted. Diagnosis criteria included positive peripheral blood smear (PBS), seroconversion of IgG, and a two-fold increase in IgG titer (laboratory criteria); and clinical findings, epidemiological exposure, and at least one positive IgG test (clinical-epidemiological criteria). In-house conventional polymerase chain reaction (PCR) was performed on 33 samples. All patients were treated with benznidazole. After 4.5 years, IgG levels were reassessed in 26 individuals. The mean age was 33.6 years, with no gender difference. The mean incubation period was 13.8 days, and the mean between symptom onset and treatment was 16.6 days. The most common symptoms were fever and lymphadenopathy (90%). Diagnostic success rates were 66.6% (laboratory criteria), 23% (clinical-epidemiological criteria), and 10.2% (high clinical suspicion despite negative tests). Test positivity rates were 69.7% (PBS), 91.4% (serology), and 100% (PCR). There were no deaths. Serological cure was achieved in 34.6% of cases, and IgG titers decreased in 15.3%.

Conclusions

We encountered several barriers in managing ACD, including population vulnerability, reliance on outdated diagnostic techniques, lack of standardized molecular biology methods, and limited therapeutic options. This report underscores the importance of rapid surveillance and early treatment to prevent fatalities. We recommend the standardization of conventional PCR in diagnostic routines.

References

---

## [Editor Report · Decision Letter 2]

24 Jul 2024

Dear Prof Alves Simoes Neto,

We are pleased to inform you that your manuscript 'Oral Chagas disease outbreak by bacaba juice ingestion: a century after discovery by Carlos Chagas, the disease is still hard to manage.' has been provisionally accepted for publication in PLOS Neglected Tropical Diseases.

Best regards,

Andrés F. Henao-Martínez, M.D.

Academic Editor

Susan Madison-Antenucci

Section Editor

---

## [Editor Report · Acceptance letter]

4 Sep 2024

Dear Prof Simões-Neto,

We are delighted to inform you that your manuscript, "Oral Chagas disease outbreak by bacaba juice ingestion: a century after discovery by Carlos Chagas, the disease is still hard to manage.," has been formally accepted for publication in PLOS Neglected Tropical Diseases.

Best regards,

Shaden Kamhawi

co-Editor-in-Chief

Paul Brindley

co-Editor-in-Chief
